# Think Deep, Not Just Long:
# Measuring LLM Reasoning Effort via Deep-Thinking Tokens

**Wei-Lin Chen** [1 2 *]  **Liqian Peng** [2]  **Tian Tan** [2]  **Chao Zhao** [2]  **Jianhang Chen** [2]  **Ziqian Lin** [2]
**Alec Go** [2]  **Yu Meng** [1]

## Abstract

Large language models (LLMs) have demonstrated impressive reasoning capabilities by scaling test-time compute via long Chain-of-Thought (CoT). However, recent findings suggest that raw token counts are unreliable proxies for reasoning quality: increased generation length does not consistently correlate with accuracy and may instead signal "overthinking," leading to performance degradation. In this work, we quantify inference-time effort by identifying *deep-thinking tokens*—tokens where internal predictions undergo significant revisions in deeper model layers prior to convergence. Across four challenging mathematical and scientific benchmarks (AIME 24/25, HMMT 25, and GPQA-diamond) and a diverse set of reasoning-focused models (GPT-OSS, DeepSeek-R1, and Qwen3), we show that *deep-thinking ratio* (the proportion of deep-thinking tokens in a generated sequence) exhibits a robust and consistently positive correlation with accuracy, substantially outperforming both length-based and confidence-based baselines. Leveraging this insight, we introduce Think@$n$, a test-time scaling strategy that prioritizes samples with high deep-thinking ratios. We demonstrate that Think@$n$ matches or exceeds standard self-consistency performance while significantly reducing inference costs by enabling the early rejection of unpromising generations based on short prefixes.

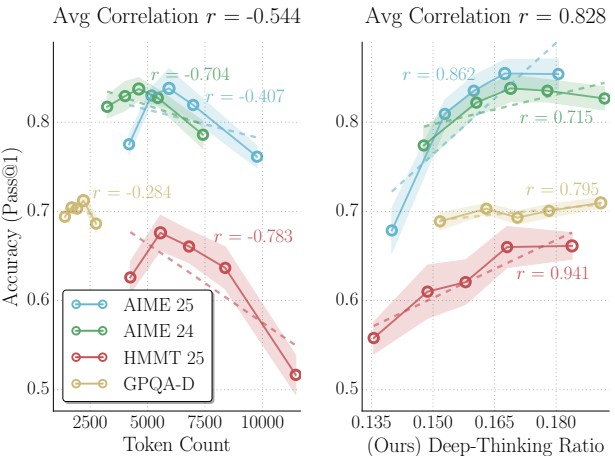

*Figure 1.* Comparison of correlations between accuracy and proxies for thinking effort. The plots illustrate the relationship between model performance and two inference-time measures of thinking effort on GPT-OSS-120B-*medium* across AIME 2024/2025, HMMT 2025, and GPQA-Diamond. *(Left)* Output token count exhibits a moderate negative correlation (average $r = -0.544$), suggesting that output length is an unreliable indicator of performance. *(Right)* In contrast, our proposed deep-thinking ratio demonstrates a strong positive correlation with accuracy (average $r = 0.828$).

## 1. Introduction

Large language models (LLMs) have achieved remarkable reasoning capabilities by generating explicit thought traces, most notably through the Chain-of-Thought (CoT) paradigm (Wei et al., 2022). Prior works have shown that increasing the number of reasoning tokens generated can generally boost task performance (Jaech et al., 2024; Guo et al., 2025; Anthropic, 2025a;b; OpenAI, 2025; Yang et al., 2025a; Team et al., 2025; Zhong et al., 2024), motivating methods that encourage longer and more elaborate thinking traces (Muennighoff et al., 2025; Balachandran et al., 2025; Yang et al., 2025b).

However, a growing body of evidence suggests that token counts are unreliable indicators of model performance during inference, as longer reasoning does not consistently translate into higher accuracy (Wu et al., 2025; Aggarwal

---

[*]Work done as a student researcher at Google [1]University of Virginia [2]Google. Correspondence to: Wei-Lin Chen <wlchen@virginia.edu>, Liqian Peng <liqianp@google.com>.

*Proceedings of the 43rd International Conference on Machine Learning*, Seoul, South Korea. PMLR 306, 2026. Copyright 2026 by the author(s).

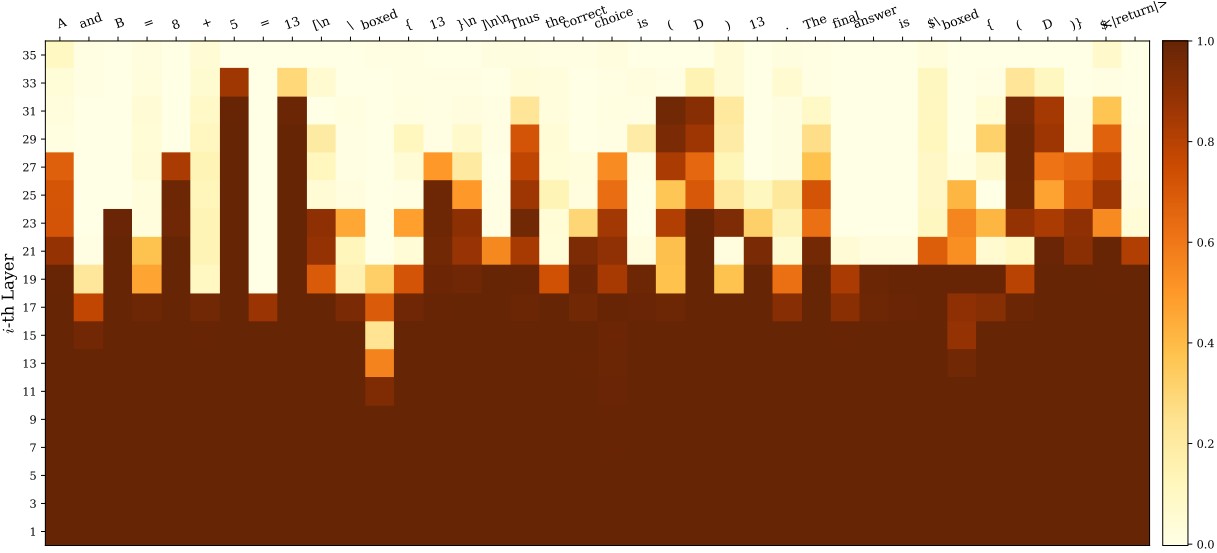

*Figure 2.* Heatmap of thought: We plot the Jensen–Shannon divergence (JSD) values between the distributions of the last (36th) layer and intermediate layers for an answer sequence from GPT-OSS-120B-*high*. Functional and templated words (*e.g.*, "and", "is", "boxed", "<|return|>") often converge at relatively shallow layers; Completions after operators (*e.g.*, "+", "=") and answer tokens/symbols (*e.g.*, "13", "(D)") do not settle until deeper layers. Interestingly, the answer token "13" gradually surfaces in earlier layers after its first appearance.

et al., 2025; Sui et al., 2025; Su et al., 2025). Empirical studies reveal inverted-U relationships between CoT length and performance (Wu et al., 2025), as well as inverse-scaling behaviors in which longer reasoning traces systematically degrade performance (Gema et al., 2025). Excessive reasoning may reflect overthinking, wherein models amplify flawed heuristics or fixate on irrelevant details (Feng et al., 2025). Consequently, relying on length as a metric for reasoning quality not only encourages verbosity over clarity but also wastes computational resources on uninformative tokens. Though recent work has attempted to assess the semantic structure of CoTs (*e.g.*, by representing reasoning traces as graphs), such approaches often rely on costly auxiliary parsing or external annotations (Feng et al., 2025). Addressing these limitations requires more principled and efficient methods for measuring thinking effort that can distinguish effective reasoning from uninformative generation.

In this work, we introduce *deep-thinking ratio* (DTR) as a direct measure of inference-time thinking effort. Instead of relying on surface-level features like output length, we focus on how individual tokens are produced internally. We posit that when a token prediction stabilizes in early layers, subsequent depth-wise modifications entail relatively low computational effort, resembling *less thinking*. In contrast, token predictions that undergo sustained revision in deeper layers before converging reflect *greater thinking* (Chuang et al., 2024). We operationalize this idea by projecting intermediate-layer hidden states into the vocabulary space and comparing each layer's prediction distribution to the

final-layer distribution. Tokens whose distributions do not converge until deeper layers are identified as *deep-thinking tokens*. By counting the proportion of deep-thinking tokens in a generated sequence, we obtain DTR, which provides a simple, mechanistically grounded measure of thinking effort, requiring neither task-specific heuristics nor external structural annotations.

Across four challenging mathematical and scientific reasoning benchmarks—AIME 2024, AIME 2025, HMMT 2025, and GPQA (Art of Problem Solving, 2024a;b; 2025a;b; HMMT, 2025; Rein et al., 2024)—and a range of reasoning-focused language models, including GPT-OSS, DeepSeek-R1, and Qwen3 families (OpenAI et al., 2025; Guo et al., 2025; Yang et al., 2025a), we demonstrate that measuring deep-thinking tokens yields strong correlations with task accuracy. The achieved correlation is substantially higher than those obtained using length-based or confidence-based baselines. Furthermore, we show that deep-thinking tokens can be leveraged for parallel inference scaling, where preferentially selecting and aggregating responses with higher DTR achieves performance comparable or better than standard consensus-based methods, while requiring only half the compute cost. Our contributions are summarized as follows:

- We introduce *deep-thinking ratio* (DTR)—a measure that counts the ratio of *deep-thinking tokens* in a sequence whose predictions undergo sustained revision in deeper layers before converging—as a new lens for characterizing inference-time thinking effort.

- We empirically show that, across multiple reasoning benchmarks and model families, DTR of a generated sequence exhibits strong positive correlations with task accuracy, outperforming length-based and confidence-based baselines significantly.

- We introduce Think@$n$, a test-time scaling strategy that preferentially selects and aggregates samples with higher DTR. By early halting unpromising generations based on DTR estimated from short prefixes, Think@$n$ matches or surpasses standard self-consistency with approximately half the inference cost.

## 2. Measuring Deep-Thinking Ratio

### 2.1. Preliminaries

We consider an autoregressive language model $f_\theta$ composed of $L$ transformer layers, hidden dimension $d$, and vocabulary $V$. Given a prefix sequence $y_{<t}$, the forward pass at generation step $t$ produces a sequence of residual stream states $\{h_{t,l}\}_{l=1}^{L}$, where $h_{t,l} \in \mathbb{R}^d$ denotes the hidden state after layer $l$. The final-layer output $h_{t,L}$ is projected by the language modeling head (*i.e.*, the unembedding matrix) $W_U \in \mathbb{R}^{|V| \times d}$ to produce logits over the vocabulary.

Prior research on early exiting (Teerapittayanon et al., 2016; Elbayad et al., 2020; Schuster et al., 2022; Din et al., 2024; Belrose et al., 2023) has demonstrated that, without specialized auxiliary training, applying the language modeling head directly to intermediate-layer hidden states effectively yields meaningful predictive distributions (Nostalgebraist, 2020; Kao et al., 2020). Building on this line of works, we project intermediate-layer hidden states into the vocabulary space using the same unembedding matrix $W_U$. For each intermediate layer $l \in \{1, \ldots, L-1\}$, we compute the logit vector $z_{t,l}$ and probability distribution $p_{t,l}$ as

$$p_{t,l} = \text{softmax}(z_{t,l}), \quad z_{t,l} = W_U h_{t,l} \qquad (1)$$

The model's final-layer distribution is denoted by $p_{t,L}$.

### 2.2. Deep-Thinking Tokens

We posit that inference-time thinking effort for a token manifests as the continued evolution of predictive distributions (*i.e.*, $p_{t,l}$) across LM layers. Tokens with earlier distributional stabilization correspond to less additional thinking, while those having later stabilization correspond to needing more extended internal thinking. In other words, simple tokens stabilize early with shallow computation, whereas difficult tokens requiring more thinking exhibit distributional shifts in deeper layers with more computation. To illustrate this, we show a motivation example on answering a GQPA (Rein et al., 2024) question in Figure 2.

To quantify this behavior, we measure how long a token's

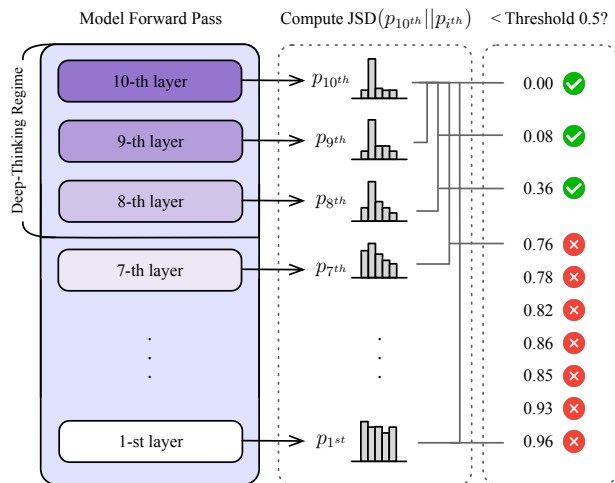

*Figure 3.* Illustration of our method of identifying deep-thinking tokens. Suppose a model with 10 layers, by setting the depth fraction $\rho = 0.8$, the token is successfully classified as a deep-thinking token at generation step $t$ since its JSD with the final-layer distribution first fall below the threshold $g$ only until it reaches the late-settling regime.

predictive distribution continues to change before *settling*, operationalized as the layer at which the intermediate distribution becomes sufficiently close to the final-layer distribution. Specifically, for each generation step $t$ and layer $l$, we compute the Jensen–Shannon divergence (JSD) between the intermediate-layer distribution $p_{t,l}$ and the final-layer distribution $p_{t,L}$:

$$
\begin{aligned}
D_{t,l} &\coloneqq \text{JSD}(p_{t,L} \,\|\, p_{t,l}) \\
&= H\left(\frac{p_{t,L} + p_{t,l}}{2}\right) - \tfrac{1}{2}H(p_{t,L}) - \tfrac{1}{2}H(p_{t,l}),
\end{aligned} \qquad (2)
$$

where $H(\cdot)$ denotes Shannon entropy. By construction, $D_{t,L} = 0$. A trajectory $l \mapsto D_{t,l}$ that approaches zero only at later layers indicates prolonged distributional revision (think more), whereas early convergence indicates that the model settles on its final prediction with fewer subsequent updates (think less). We employ JSD due to its symmetry and boundedness, following (Chuang et al., 2024). We explore other distance metrics in Section B.

To enforce a strict notion of settling, we compute:

$$\bar{D}_{t,l} = \min_{j \leq l} D_{t,j}. \qquad (3)$$

We define the settling depth $c_t$ as the first layer at which $\bar{D}_{t,l}$ falls below a fixed threshold $g$:

$$c_t = \min\left\{l \in \{1, \ldots, L\} : \bar{D}_{t,l} \leq g\right\}. \qquad (4)$$

**Algorithm 1** Computing Deep-Thinking Ratio (DTR)

**Input** : Autoregressive LM $f_\theta$ with $L$ layers and unembedding matrix $W_U$; Input prompt $x$; Threshold $g$; Depth fraction $\rho$

**Output** : $\mathrm{DTR}(S)$ of the generated sequence $S$

$C \leftarrow 0$;          // deep thinking token count
$S \leftarrow \emptyset$;               // generated sequence
$y_t \leftarrow$ [BOS] ; // initialize with start token
**while** $y_t \neq$ [EOS] **do**
    Sample $y_t \sim p_{t,L}(f_\theta(\cdot \mid x, S))$ $S \leftarrow (S, y_t)$ **for** $l \leftarrow 1$
    **to** $L$ **do**
     |   $p_{t,l} \leftarrow \mathrm{softmax}(W_U h_{t,l})$ $D_{t,l} \leftarrow \mathrm{JSD}(p_{t,L}, p_{t,l})$
    **end**
    $c_t \leftarrow \min\{l : \min_{j \leq l} D_{t,j} \leq g\}$ **if** $c_t \geq \lceil \rho \times L \rceil$ **then**
     |   $C \leftarrow C + 1$
    **end**
**end**
**return** $C/|S|$

We then define a deep-thinking regime using a depth fraction $\rho \in (0, 1)$, with

$$\mathcal{L}_{\text{deep-thinking}} = \{l : l \geq \lceil \rho \times L \rceil\}. \quad (5)$$

A token is classified as a deep-thinking token (*i.e.*, requiring more layer computations and more thinking effort to become sufficiently close to the final-layer distribution) if $c_t \in \mathcal{L}_{\text{deep-thinking}}$. An illustration is shown in Figure 3.

Finally, for a generated sequence $S$ of length $T$, we define the deep-thinking ratio, $\mathrm{DTR}(S)$, for the sequence as the proportion of tokens that settle in the deep-thinking regime:

$$\mathrm{DTR}(S) = \frac{1}{T} \sum_{t=1}^{T} \mathbb{1}[c_t \in \mathcal{L}_{\text{deep-thinking}}]. \quad (6)$$

A higher DTR indicates that a larger fraction of tokens undergo extended computation for distributional revision before stabilizing. We note that our proposed method does not imply that early-settling tokens are suboptimal; rather, it provides a depth-wise characterization of inference-time thinking effort that complements the surface-level token length measure. We show the overall algorithm of DTR in Algorithm 1. We also provide qualitative examples in Section F.

## 3. Deep-Thinking Ratio Reflects Task Accuracy More Reliably

We empirically evaluate whether our distributional distance-based measurement provides a more faithful and robust characterization of inference-time thinking effort than surface-level, length-based proxies (*i.e.*, token counts).

**Models.** We evaluate eight variants of reasoning LLMs from three model families: GPT-OSS-20B (with low, medium, and high reasoning levels) and GPT-OSS-120B (with low, medium, and high reasoning levels) (OpenAI et al., 2025), DeepSeek-R1-70B (Guo et al., 2025),[1] and Qwen3-30B-Thinking (Yang et al., 2025a). These models are known for their strong, long CoT capability in mathematical and complex reasoning, and span multiple parametric scales for comprehensive coverage.

**Tasks.** We focus on reasoning-intensive benchmarks where scaling CoT-style computation at inference time plays a central role. We adopt four benchmarks widely used in recent evaluations of LLM reasoning capabilities (xAI, 2025; OpenAI, 2025; Balunovic et al., 2025), including three competition-level mathematical problem sets, AIME 2024 (Art of Problem Solving, 2024a;b), AIME 2025 (Art of Problem Solving, 2025a;b), and HMMT 2025 (HMMT, 2025), as well as the diamond set of GPQA (Rein et al., 2024), which consists of challenging graduate-level scientific questions.

**Decoding settings.** Following (Gema et al., 2025), we prompt models to reason step by step using a fixed, neutral instruction, without specifying a reasoning budget or explicitly encouraging longer deliberation. This setup allows each model to naturally allocate inference-time computation on a per-instance basis, avoiding confounds introduced by externally imposed token budgets or budget-conditioning prompts. Following standard practice in natural overthinking analyses (Gema et al., 2025), we sample multiple responses for each question (25 responses per question in our experiments). Across these samples, models naturally exhibit variation in reasoning length and internal computation patterns. We use the developer recommended sampling parameters for all tested models: temperature=1.0 and top $p$=1.0 for GPT-OSS series; temperature=0.6 and top $p$= 0.95 for DeepSeek-R1-70B and Qwen-3-30B-Thinking.

For each sampled response, we record intermediate-layer hidden states, obtain their projected probability distribution, and compute DTR as described in Section 2. We uniformly set the settling threshold $g = 0.5$ and the depth fraction $\rho = 0.85$ to define the deep-thinking regime. We also analyze with different values and the results are provided in Section 3.2. The reported statistics are averaged over 30 random seeds across decoding runs.

---

[1]For brevity, we refer DeepSeek-R1-70B to Llama-3.3-70B-Instruct distilled with DeepSeek-R1 generated samples (https://huggingface.co/deepseek-ai/DeepSeek-R1-Distill-Llama-70B).

## 3.1. Results

To quantify the relationship between inference-time thinking effort and task performance, we measure the association between thinking effort scores and answer accuracy by computing Pearson correlation coefficient. Specifically, we conduct a binned analysis following (Gema et al., 2025) by partitioning sampled sequences into quantile bins (*i.e.*, 5 bins) based on their DTR (Equation (6)) and computing the average accuracy within each bin.

We compare deep-thinking token measurement against the following baselines, including length-based proxies and confidence-based approaches, which are also commonly adopted to assess generation quality.

**Token count.**   The total number of tokens generated in the model's output reasoning traces. This measure is widely framed as a direct proxy for test-time compute, and underlies many empirical studies of inference-time scaling (Jaech et al., 2024; Guo et al., 2025; Anthropic, 2025a;b; OpenAI, 2025; Yang et al., 2025a; Team et al., 2025; Zhong et al., 2024).

**Reverse token count.**   As a complementary baseline, we additionally consider reverse token count, defined as the negative of the total number of generated tokens for each response. This transformation is included to account for the frequently observed inverse relationship between reasoning length and accuracy in LLM overthinking (Wu et al., 2025; Gema et al., 2025).

**Log probability.**   Following the notation in Section 2, let a generated sequence $S = (y_1, \ldots, y_T)$. At generation step $t$, the model's output prediction distribution (at final-layer $L$) over the vocabulary $\mathcal{V}$ is denoted by $p_{t,L}(\cdot)$. We compute the average log-probability of the sampled tokens:

$$\text{LogProb}(S) \;=\; \frac{1}{T}\sum_{t=1}^{T}\log p_{t,L}(y_t) \tag{7}$$

Higher values indicate that the model assigns higher likelihood to its own generation and are commonly interpreted as higher confidence.

**Negative perplexity.**   Perplexity is defined as the exponentiated negative average log-probability:

$$\text{PPL}(S) \;=\; \exp\!\left(-\frac{1}{T}\sum_{t=1}^{T}\log p_{t,L}(y_t)\right) \tag{8}$$

We report negative perplexity $-\text{PPL}(S)$ so that larger values correspond to higher confidence.

**Negative entropy.**   To incorporate information from the full prediction distribution over $\mathcal{V}$ rather than only the sampled token, we compute the average entropy:

$$\text{Ent}(S) \;=\; \frac{1}{T}\sum_{t=1}^{T}H(p_{t,L}),$$
$$H(p_{t,L}) = -\sum_{v\in\mathcal{V}} p_{t,L}(v)\log p_{t,L}(v) \tag{9}$$

We report negative entropy $-\text{Ent}(S)$, where larger values indicate more peaked distributions and thus greater model confidence.

**Self-Certainty.**   We also include Self-Certainty (Kang et al., 2025), a distributional confidence metric based on the idea that higher confidence corresponds to prediction distributions that are further from the uniform distribution $u$, which represents maximum uncertainty. Formally, self-certainty is defined as the average Kullback-Leibler (KL) divergence between $u(v) = 1/|\mathcal{V}|$ and $p_{t,L}$:

$$\text{Self-Certainty}(S) \;=\; \frac{1}{T}\sum_{t=1}^{T}\text{KL}(u \,\|\, p_{t,L})$$
$$= -\frac{1}{T|\mathcal{V}|}\sum_{t=1}^{T}\sum_{v\in\mathcal{V}}\log\big(|\mathcal{V}|\,p_{t,L}(v)\big) \tag{10}$$

For all baselines, correlations are computed using the same protocol, where sequences are ranked and binned by token count (or its negation) or confidence scores.

Table 1 reports the correlation between task accuracy and different measurments, across eight model variants and four benchmarks. As observed, measuring sequences with token count exhibits notable oranged-colored values ($r < 0$), with mean $r = -0.59$. This indicates that longer generations are more associated with lower performance, aligning with recent reports of inverse scaling and overthinking. Extended reasoning traces could be symptomatic of redundant, misguided, or error-amplifying deliberation. The results underscore the unreliability of using surface-level length feature as proxy for effective problem solving. Reversing token count yields a positive correlation of identical magnitude. However, the improvement is purely post hoc, reflecting the empirical regularity in regimes where shorter responses are more accurate. As such, reverse token count only serve as a statistical adjustment, rather than capture principled notion of computation or thinking effort.

Compared to token count measure, confidence-based measures (log probability, negative perplexity, negative entropy, and self-certainty) exhibit moderately positive correlations with mean $r = 0.219 \sim 0.605$, as reflected by the predominance of green-colored values. This indicates that

*Table 1.* Pearson correlations between task accuracy and different inference-time measures, including length-based and confidence-based baselines, across eight model variants and four reasoning benchmarks. Correlation values are color-coded: strong positive correlations $(0.5 \sim 1)$ are shown in dark green, weak positive correlations $(0 \sim 0.5)$ in light green, weak negative correlations $(-0.5 \sim 0)$ in light orange, and strong negative correlations $(-1 \sim -0.5)$ in dark orange.

| | Token Length | Reverse Token Length | Log Probability | Negative Perplexity | Negative Entropy | Self-Certainty | **DTR (Ours)** |
|---|---|---|---|---|---|---|---|
| | | | *AIME 2025* | | | | |
| OSS-120B-low | 0.504 | -0.504 | 0.872 | 0.453 | 0.863 | 0.803 | 0.930 |
| OSS-120B-medium | -0.365 | 0.365 | 0.817 | 0.246 | 0.822 | 0.815 | 0.862 |
| OSS-120B-high | -0.961 | 0.961 | 0.705 | 0.552 | 0.711 | 0.728 | 0.796 |
| OSS-20B-low | -0.689 | 0.689 | 0.579 | 0.849 | 0.665 | 0.275 | 0.373 |
| OSS-20B-medium | -0.757 | 0.757 | 0.616 | -0.677 | 0.637 | 0.097 | 0.161 |
| OSS-20B-high | -0.385 | 0.385 | 0.455 | -0.795 | 0.550 | 0.489 | 0.610 |
| DeepSeek-R1-70B | -0.973 | 0.973 | 0.961 | 0.955 | 0.946 | 0.899 | 0.974 |
| Qwen3-30B-Thinking | -0.663 | 0.663 | -0.008 | -0.035 | 0.154 | 0.828 | 0.855 |
| | | | *AIME 2024* | | | | |
| OSS-120B-low | -0.166 | 0.166 | 0.897 | 0.682 | 0.869 | 0.741 | 0.840 |
| OSS-120B-medium | -0.680 | 0.680 | 0.795 | -0.293 | 0.908 | 0.924 | 0.715 |
| OSS-120B-high | -0.755 | 0.755 | 0.700 | -0.275 | 0.593 | 0.654 | 0.905 |
| OSS-20B-low | -0.655 | 0.655 | 0.548 | -0.342 | 0.667 | 0.584 | 0.730 |
| OSS-20B-medium | -0.827 | 0.827 | 0.195 | -0.150 | 0.440 | 0.252 | -0.192 |
| OSS-20B-high | -0.989 | 0.989 | 0.809 | 0.262 | 0.921 | 0.855 | 0.824 |
| DeepSeek-R1-70B | -0.987 | 0.987 | -0.037 | 0.223 | 0.067 | 0.287 | 0.430 |
| Qwen3-30B-Thinking | -0.869 | 0.869 | -0.857 | -0.720 | -0.680 | -0.246 | -0.657 |
| | | | *GPQA-Diamond* | | | | |
| OSS-120B-low | 0.682 | -0.682 | 0.984 | 0.172 | 0.995 | 0.996 | 0.976 |
| OSS-120B-medium | -0.340 | 0.340 | 0.973 | 0.316 | 0.985 | 0.981 | 0.795 |
| OSS-120B-high | -0.970 | 0.970 | 0.854 | 0.501 | 0.813 | 0.885 | 0.845 |
| OSS-20B-low | -0.602 | 0.602 | 0.984 | 0.235 | 0.991 | 0.917 | 0.935 |
| OSS-20B-medium | -0.847 | 0.847 | 0.914 | 0.468 | 0.911 | 0.889 | 0.718 |
| OSS-20B-high | -0.794 | 0.794 | 0.879 | 0.461 | 0.902 | 0.915 | 0.962 |
| DeepSeek-R1-70B | -0.930 | 0.930 | 0.068 | -0.133 | -0.165 | -0.532 | 0.885 |
| Qwen3-30B-Thinking | -0.634 | 0.634 | 0.589 | 0.865 | 0.711 | 0.943 | 0.828 |
| | | | *HMMT 2025* | | | | |
| OSS-120B-low | 0.871 | -0.871 | 0.761 | 0.629 | 0.695 | 0.884 | 0.305 |
| OSS-120B-medium | -0.793 | 0.793 | 0.706 | 0.045 | 0.618 | 0.631 | 0.941 |
| OSS-120B-high | -0.967 | 0.967 | 0.750 | 0.503 | 0.728 | 0.754 | 0.972 |
| OSS-20B-low | -0.634 | 0.634 | -0.695 | 0.549 | -0.359 | -0.489 | 0.689 |
| OSS-20B-medium | -0.668 | 0.668 | 0.447 | 0.336 | 0.424 | 0.331 | 0.247 |
| OSS-20B-high | -0.352 | 0.352 | 0.537 | 0.994 | 0.831 | 0.628 | 0.932 |
| DeepSeek-R1-70B | -0.866 | 0.866 | 0.879 | 0.889 | 0.858 | 0.905 | 0.902 |
| Qwen3-30B-Thinking | -0.950 | 0.950 | -0.803 | -0.762 | -0.801 | 0.745 | 0.911 |
| **Average** | -0.594 | 0.594 | 0.527 | 0.219 | 0.571 | 0.605 | **0.687** |

model confidence captures partial information about correctness. However, their behavior is relatively heterogeneous across models and benchmarks: while certain configurations achieve strong positive correlations, others deteriorate to weak or even negative associations. This inconsistency suggests that confidence signals might conflate other factors like overconfidence, and therefore do not reliably reflect inference-time compute effort or problem solving effectiveness.

In contrast, our proposed measurement of DTR demonstrates the strongest and most stable relationship with task performance, achieving the highest average correlation of $r = 0.687$, outperforming both reverse token count and Self-Certainty, the best-performing baselines among confidence-based approaches. Overall, DTR remains positive across models and benchmarks, exhibiting the fewest orange-colored values (2 out of the 32 model–benchmark settings tested). Collectively, the results show that computing DTR over output sequences provides a more faithful and robust characterization of successful reasoning outcomes than token volume alone or confidence-based alternatives.

### 3.2. Effect of Settling Thresholds and Depth Fractions

We conduct an analysis to understand how our two key hyper-parameters—the settling threshold $g$ and the depth fraction $\rho$ defining the deep-thinking regime—affect the measured thinking effort and its correlation with task performance. Figure 4 illustrates the accuracy profiles across vary-

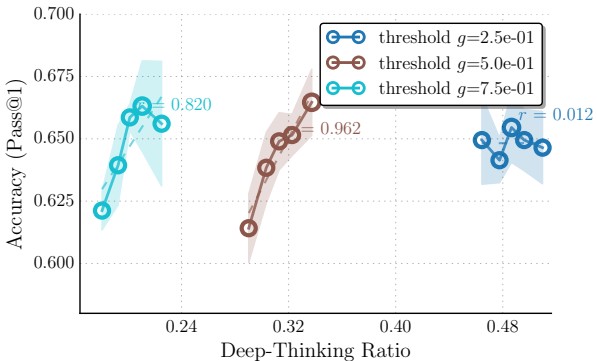

*(a) Effect of different settling threshold $g$.*

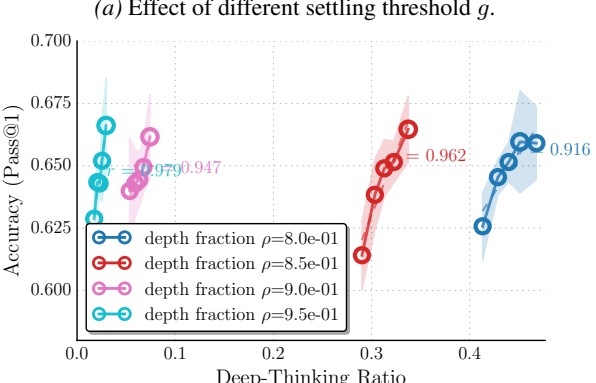

*(b) Effect of different depth fraction $\rho$.*

*Figure 4.* Effect of hyper-parameters on thinking effort measurement and accuracy profiles. We analyze the impact of hyper-parameters by sweeping different settling threshold $g$ and depth fraction $\rho$. (a) Varying $g$ has more impacts the correlation; a permissive threshold ($g = 0.25$) yields flatter trends, whereas $g = 0.5$ provides the most robust positive signal. (b) Varying $\rho$ shifts the range of thinking effort scores but maintains overall consistent positive slopes. Overall, stricter criteria (higher $g$, lower $\rho$) reduce the range of DTR, with $(g, \rho) = (0.5, 0.85)$ offering an ideal balance between stability and correlation.

ing deep-thinking ratios, derived by $g \in \{0.25, 0.5, 0.75\}$ and $\rho \in \{0.8, 0.85, 0.9, 0.95\}$. We set $\rho$ fixed to 0.85, when sweeping $g$, and $g$ fixed to 0.5 when sweeping $\rho$. We report results on GPQA-D using GPT-OSS-20B with reasoning level high.

We conclude the following observations: *(1)* the magnitude of the measured sequence-level thinking effort is directly influenced by the strictness of these parameters. Specifically, both Figures 4a and 4b show that imposing stricter criteria—a higher settling threshold $g$ or a lower depth fraction $\rho$—results in a reduction of the deep-thinking ratio. This is mechanistically consistent: a higher $g$ requires the intermediate states to be distributionally far to the final output until reaching deeper layers in the late regime to be considered settle; while a lower $\rho$ restricts the definition of the deep-thinking regime to a narrower band of deeper layers. Both conditions naturally filter out more candidates, resulting in

fewer tokens being classified as deep-thinking tokens and consequently a lower range of overall deep-thinking ratios.

*(2)* The settling threshold $g$ has a more pronounced impact on the correlation between thinking effort and accuracy than the depth fraction $\rho$. As shown in Figure 4b, varying $\rho$ shifts the range of deep-thinking ratios due to varying strictness but maintains a consistent, positive slope across all settings, indicating that the metric is relatively robust to the specific definition of the late layers. In contrast, Figure 4a reveals that the choice of $g$ has more impact on measured results: a softer threshold of $g = 0.25$ yields a flatter trend with lower correlation value, suggesting that it may be overly permissive, including tokens with less computational efforts and diminishing the measurement's ability to distinguish high-quality trajectory. Conversely, thresholds of $g = 0.5$ and $g = 0.75$ exhibit more robust positive correlations reflecting the accuracy.

*(3)* Overall, we can see that when the criteria are overly restrictive ($g = 0.75$ and $\rho \in \{0.9, 0.95\}$), the trends, while still maintaining positive correlations, appears to be slightly more unstable due to the potential filtering of informative high computational tokens. Among the tested configurations, $(g, \rho) = (0.5, 0.85)$ strikes an ideal balance, yielding a reliable trend with high correlation values.

## 4. Deep-Thinking Tokens Enable Efficient Test-Time Scaling

Repeated sampling is a popular strategy for scaling test-time compute, in parallel to generating long CoT (Brown et al., 2024; Gupta & Srikumar, 2025; Saad-Falcon et al., 2024; Stroebl et al., 2026; Saad-Falcon et al., 2025). It improves accuracy by aggregating multiple independently generated samples per problem at the cost of increased inference budget. In this section, we explore whether our proposed DTR measure can be leveraged to preferentially select and aggregate higher-quality samples towards better performance.

**Experimental setups.** We follow the best-of-n (BoN) evaluation protocol commonly adopted in recent test-time scaling studies (Fu et al., 2026). For each problem, we sample $n$ responses using identical decoding settings, and compare the following aggregation methods: **Cons@$n$:** Standard self-consistency (Wang et al., 2023), which performs majority voting over all $n$ sampled responses; **Mean@$n$:** The average accuracy of all the $n$ samples, reflecting a baseline of no preferential aggregation; **Long@$n$** and **Short@$n$:** Majority voting over the longest/shortest $\eta$ percent of the $n$ samples, ranked by token count (Hassid et al., 2025; Agarwal et al., 2025). **Self-Certainty@$n$:** Majority voting over the highest-scoring $\eta$ percent of the $n$ samples, ranked by Self-Certainty score (the best-performing baseline in Section 3); **Think@$n$:** Majority voting over the highest-scoring

*Table 2.* Comparison of task accuracy and average inference cost (k tokens) under different aggregation methods, across four reasoning benchmarks. The reported cost reductions ($\Delta\%$) are shown relative to Cons@$n$. Think@$n$ achieves the best overall performance while reducing inference cost by approximately 50%. Methods with † adopt a prefix length of 50 to determine early stopping.

| Method | AIME 25 | | AIME 24 | | HMMT 25 | | GPQA-D | |
|---|---|---|---|---|---|---|---|---|
| | **Acc** | **Cost ($\Delta\%$)** | **Acc** | **Cost ($\Delta\%$)** | **Acc** | **Cost ($\Delta\%$)** | **Acc** | **Cost ($\Delta\%$)** |
| | | | | *OSS-120B-medium* | | | | |
| Cons@$n$ | 92.7 | 307.6 (–) | 92.7 | 235.1 (–) | 80.0 | 355.6 (–) | 73.8 | 93.5 (–) |
| Mean@$n$ | 80.0 | 307.6 (–) | 81.6 | 235.1 (–) | 62.6 | 355.6 (–) | 69.9 | 93.5 (–) |
| Long@$n$ | 86.7 | 307.6 (–) | 86.7 | 235.1 (–) | 73.3 | 355.6 (–) | 73.2 | 93.5 (–) |
| Short@$n$ | 87.3 | 255.7 (-17%) | 88.0 | 200.9 (-15%) | 77.3 | 290.4 (-18%) | 73.3 | 84.4 (-10%) |
| Self-Certainty@$n$† | 87.3 | 150.6 (-51%) | 91.3 | 119.3 (-49%) | 78.0 | 177.0 (-50%) | **76.0** | 47.9 (-49%) |
| Think@$n$† | **94.7** | 155.4 (-49%) | **93.3** | 121.3 (-48%) | **80.0** | 181.9 (-49%) | 74.7 | 48.8 (-48%) |
| | | | | *Qwen3-4B-Thinking* | | | | |
| Cons@$n$ | 86.7 | 1073.1 (–) | 93.3 | 950.1 (–) | 63.3 | 1275.7 (–) | 67.8 | 410.6 (–) |
| Mean@$n$ | 81.2 | 1073.1 (–) | 86.3 | 950.1 (–) | 55.7 | 1275.7 (–) | 66.9 | 410.6 (–) |
| Long@$n$ | 85.3 | 1073.1 (–) | 86.7 | 950.1 (–) | 52.7 | 1275.7 (–) | 66.7 | 410.6 (–) |
| Short@$n$ | 90.0 | 983.6 (-8%) | 90.0 | 871.0 (-8%) | 63.3 | 1165.7 (-9%) | 68.2 | 382.9 (-7%) |
| Self-Certainty@$n$† | 86.7 | 548.9 (-49%) | 90.0 | 480.9 (-49%) | 63.3 | 641.4 (-50%) | 68.2 | 206.6 (-50%) |
| Think@$n$† | **90.0** | 537.5 (-50%) | **93.3** | 482.2 (-49%) | **66.7** | 641.4 (-50%) | **69.7** | 206.8 (-50%) |

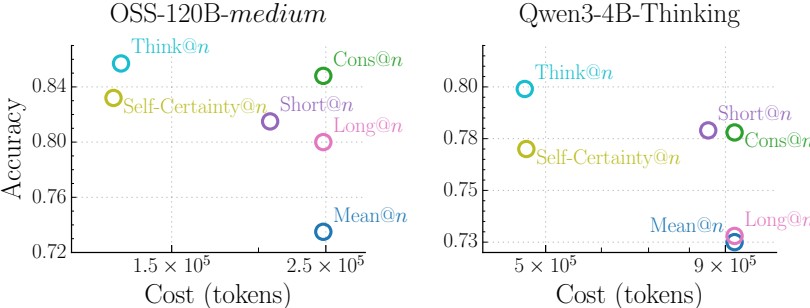

*Figure 5.* Comparison of the trade-off between task accuracy and inference cost (tokens) with different aggregation methods. Accuracy is averaged across all four datasets (AIME 24/25, HMMT 25, GPQA-D). Our Think@$n$ method achieves the best overall Pareto-optimal performance. It matches or exceeds the accuracy of Cons@$n$ with approximately half the inference cost, while Self-Certainty@$n$ is notably less efficient.

$\eta$ percent of the $n$ samples, ranked by DTR($\cdot$). All methods operate on the same pool of $n$ samples. We set $n = 48$ and $\eta = 50\%$. More analysis are provided in Section D. The results are averaged across 10 trials.

**Results.** We report the results in Table 2. To compare efficiency, we explicitly account for early stopping for Short@$n$, Self-Certainty@$n$, and Think@$n$, which aggregate only a subset of samples. Specifically, we report the average per-problem inference cost, measured as the total number of generated tokens, under the following protocols.

For Cons@$n$ and Mean@$n$, the inference cost is defined as the sum of token counts across all $n$ sampled responses= (*i.e.*, $\sum_{i=1}^{n} |S_i|$) corresponding to full decoding without early stopping. For Short@$n$, we rank samples by their length and select the shortest $\eta \times n$ samples. The inference cost is computed as the sum of the token count of the selected samples, plus an early-stopping overhead equal to

$\ell_{\text{longest\_short}} \times \eta \times n$, where $\ell_{\text{short}}$ denotes the length of the longest sample among the selected shortest subset. This term accounts for partially generated samples that are terminated once subset generation completes (*i.e.*, bounded by $\ell_{\text{longest\_short}}$). The inference cost for Long@$n$ is the same as Cons@$n$ and Mean@$n$ as it requires full decoding to select longest samples. For Think@$n$, samples are ranked by DTR, computed from a fixed prefix. Let $\ell_{\text{prefix}}$ denote the number of prefix tokens used to estimate DTR($S[: \ell_{\text{prefix}}]$). The inference cost is defined as the total token count of the top $\eta \times n$ ranked samples, plus a fixed prefix overhead of $\ell_{\text{prefix}} \times \eta \times n$, which reflects the cost of generating all candidates prior to early termination. Self-Certainty@$n$ follows the same cost computation as Think@$n$, differing only in that samples are ranked by Self-Certainty($S[: \ell_{\text{prefix}}]$) rather than DTR($S[: \ell_{\text{prefix}}]$).

Table 3 reports a preliminary ablation on AIME 25 that

*Table 3.* Impact of prefix length ($\ell_{\text{prefix}}$) on Think@$n$ performance and inference cost for AIME 2025. Using a short prefix of 50 tokens to estimate DTR outperforms using longer ones, and is comparable to full sequence (all) while providing significant cost savings. We also report Pass@1 and Cons@$n$ for reference. Subscripts denote the standard deviation across 10 trials.

|  | Accuracy | Cost (k tokens) |
|---|---|---|
| Pass@1 | $80.0_{4.2}$ | 6.4 |
| Cons@$n$ | $90.0_{2.5}$ | 307.6 |
| Think@$n$ |  |  |
| *Prefix length* |  |  |
| *50* | $94.7_{1.6}$ | 155.4 |
| *100* | $92.0_{1.6}$ | 154.1 |
| *500* | $92.7_{1.3}$ | 153.2 |
| *1000* | $92.7_{1.3}$ | 177.4 |
| *2000* | $92.0_{1.3}$ | 198.8 |
| *all* | $94.0_{0.3}$ | 307.6 |

varies $\ell_{\text{prefix}}$. We find that using only $\ell_{\text{prefix}} = 50$ tokens achieves higher accuracy than longer prefixes and matches the performance obtained using the full sequence, while significantly reducing inference cost. Accordingly, we fix $\ell_{\text{prefix}} = 50$ for all experiments in Table 2.

As shown, Cons@$n$ incurs the highest inference cost due to full decoding of every candidate, while providing a strong accuracy baseline. Mean@$n$ has the same cost as Cons@$n$ but is the worst-performing one among all methods. Under early stopping, Short@$n$ achieves modest cost savings relative to Cons@$n$, yet consistently underperforms it in accuracy. Long@$n$ exhibits further degraded performance compared to Short@$n$ without offering any cost-saving benefits. This indicates that length-based heuristics remain a coarse proxy for reasoning quality and often fail to reliably identify high-quality samples, leading to suboptimal aggregations. Self-Certainty@$n$ substantially reduces inference cost by enabling early stopping using short prefixes, but nonetheless underperforms both Cons@$n$ and Think@$n$ on three of the four evaluated benchmarks. In contrast, Think@$n$ consistently matches or exceeds the accuracy of Cons@$n$ while requiring approximately half the inference cost. The Pareto-optimal performance is most evident in the averaged results shown in Figure 5, where Think@$n$ achieves the best overall accuracy-cost trade-off. In sum, these results demonstrate that DTR provides a more informative and reliable selection signal, enabling efficient parallel scaling of inference compute.

## 5. Related Work

Test-time scaling commonly assumes that longer chains of thought (CoT) improve reasoning (Wei et al., 2022; Guo et al., 2025; Muennighoff et al., 2025), but recent work shows this heuristic is unreliable (Feng et al., 2025; Wu

et al., 2025). Gema et al. (2025) identify inverse scaling regimes where increased reasoning length reduces accuracy, while Wu et al. (2025) observe an inverted-U relationship, indicating an optimal CoT length beyond which error accumulation dominates. Several methods therefore favor concise reasoning. Hassid et al. (2025) show that shorter sampled traces are often more accurate and propose length-based voting, while Agarwal et al. (2025) select the earliest completed trace in parallel decoding to reduce tokens without accuracy loss. On the training side, Shrivastava et al. (2026) introduce Group Filtered Policy Optimization (GFPO) to suppress length inflation in RL. Our work shares the view that token count poorly reflects reasoning effort, but instead of relying on surface-level brevity, we propose an internal mechanistic signal.

Prior works has also demonstrated that intermediate representations in LMs encode meaningful predictive signals. Nostalgebraist (2020) project hidden states into vocabulary space, revealing that token predictions are formed early and refined across layers, a technique we adopt. Belrose et al. (2023) improve this with learned, layer-specific projections for clearer intermediate predictions. Beyond probing, Chuang et al. (2024) exploit layer-wise concentration of factual knowledge via contrastive decoding, while Vilas et al. (2025) use latent trajectory signals over generated traces to predict correctness. Related analyses show that early layers favor generic guesses refined in later layers (Gupta et al., 2025), and that late layers mainly perform distributional refinement (Csordás et al., 2026). These results suggest predictions often stabilize before the final layer, motivating our use of unmodified internal representations as a proxy for computational and implicitly thinking effort. An extended related work discussion is provided in Section A.

## 6. Conclusion

We introduced deep-thinking ratio (DTR) as a novel measure of inference-time reasoning effort in LLMs. By tracking depth-wise stabilization of token predictions, DTR provides a more reliable signal of effective reasoning than surface-level proxies such as token length or confidence. Building on this insight, we proposed Think@$n$, a test-time scaling strategy that leverages DTR for early selection and aggregation, achieving comparable or better performance than standard self-consistency while substantially reducing inference cost. Together, our results suggest that measuring how models think internally, rather than how long they think, is a promising direction. Future work may leverage this insight to explore how effective reasoning is characterized— shifting the focus from generating longer chains of thought to inducing deeper, more computationally intensive reasoning, and potentially enabling more reliable and efficient reasoning models.

## Acknowledgements

We thank Congchao Wang and colleagues from Google AIR for their valuable support. We also thank Yu-Min Tseng from Virginia Tech and members of Meng-Lab at UVA for their helpful discussion.

## Impact Statement

This paper presents work whose goal is to advance the field of Machine Learning. There are many potential societal consequences of our work, none which we feel must be specifically highlighted here.

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

# A. Extended Related Work

## A.1. Relationship between CoT Length and Performance

The paradigm of test-time scaling has largely operated on the assertion that allocating more computation, typically manifested as longer CoT sequences, boosts reasoning performance (Wei et al., 2022; Guo et al., 2025; Muennighoff et al., 2025). Recent empirical studies have highlighted nuances to the universality of this "longer is better" heuristic (Feng et al., 2025; Wu et al., 2025). Gema et al. (2025) identify inverse scaling regimes where increased reasoning length systematically degrades accuracy across diverse tasks, particularly when models are prone to distraction. Similarly, Wu et al. (2025) characterize the relationship between CoT length and accuracy as an "inverted-U" curve, suggesting an optimal length exists beyond which performance deteriorates due to factors like error accumulation.

Several works have proposed methods to exploit corresponding observations by favoring conciseness. Hassid et al. (2025) demonstrated that the shortest reasoning chains among sampled candidates are often the most accurate, proposing inference-time length-based voting for efficient generations. A close work by Agarwal et al. (2025) also introduced a training-free strategy that selects the first completed trace in parallel decoding, reducing token usage while maintaining accuracy. On the training side, Shrivastava et al. (2026) proposed Group Filtered Policy Optimization (GFPO) to explicitly curb length inflation in RL by rejection sampling that filters longer responses, demonstrating that models can think less without sacrificing performance. Our work aligns with these perspectives by confirming that raw token count is an unreliable proxy for effective reasoning effort, but we diverge by proposing a mechanistic internal signal rather than simply relying on surface-level brevity heuristics.

## A.2. Leveraging Internal Information in LLMs

A rich line of work has investigated how LMs internally represent and manipulate information across layers, and how internal states can be exploited. Central to this direction is the observation that intermediate representations in LMs often encode meaningful signals before reaching the final layer. Early evidence for this view was provided by Nostalgebraist (2020), which projects intermediate hidden states directly into the vocabulary space using the model's unembedding matrix—a technique we adopt in our work. The results reveal that autoregressive transformers form coarse guesses about the next token that are iteratively refined across layers. Subsequent analyses (Belrose et al., 2023) further introduce learned, layer-specific affine transformations that better align intermediate representations with the final prediction space, enabling more interpretable token predictions in shallower layers.

Beyond model probing, Chuang et al. (2024) exploits the empirical finding that factual knowledge in LMs is often more salient in particular layers. By contrasting logits from higher and lower layers, they propose a decoding method that amplifies factual signals and improves factuality. A recent work by Vilas et al. (2025) introduces latent-trajectory signals characterizing the temporal evolution of hidden states across generated reasoning traces to predict correctness. While the work examines the sequential dimension of representations, our work focuses on the depth-wise evolution of predictions across layers for individual tokens.

Complementary interpretability works also revisit how LLMs utilize depth at inference. Gupta et al. (2025) shows that early layers tend to favor high-frequency, generic token guesses, which are subsequently refined into contextually appropriate predictions. Csordás et al. (2026) suggest that later layers primarily perform fine-grained distributional refinement rather than introducing fundamentally new transformations, raising questions about the efficiency of depth utilization in modern LLMs. These findings reinforce the view that internal predictions may stabilize before the final layer, aligning with our motivations. Overall, our goal is not to modify or construct internal states to develop new methods aimed at improving model capabilities. Instead, we leverage natural, unaltered internal representations as a proxy for measuring model computational effort, which implicitly reflects thinking effort in LLMs.

# B. Comparison of Different Distance Metrics for DTR

Our method (Section 2) adopts Jensen–Shannon divergence (JSD) to quantify the discrepancy between intermediate-layer and final-layer predictions and compute DTR. Alternative notions of distance are possible. Here we explore two additional metrics: Kullback–Leibler divergence (KLD) and cosine similarity. The results are presented in Figure 6.

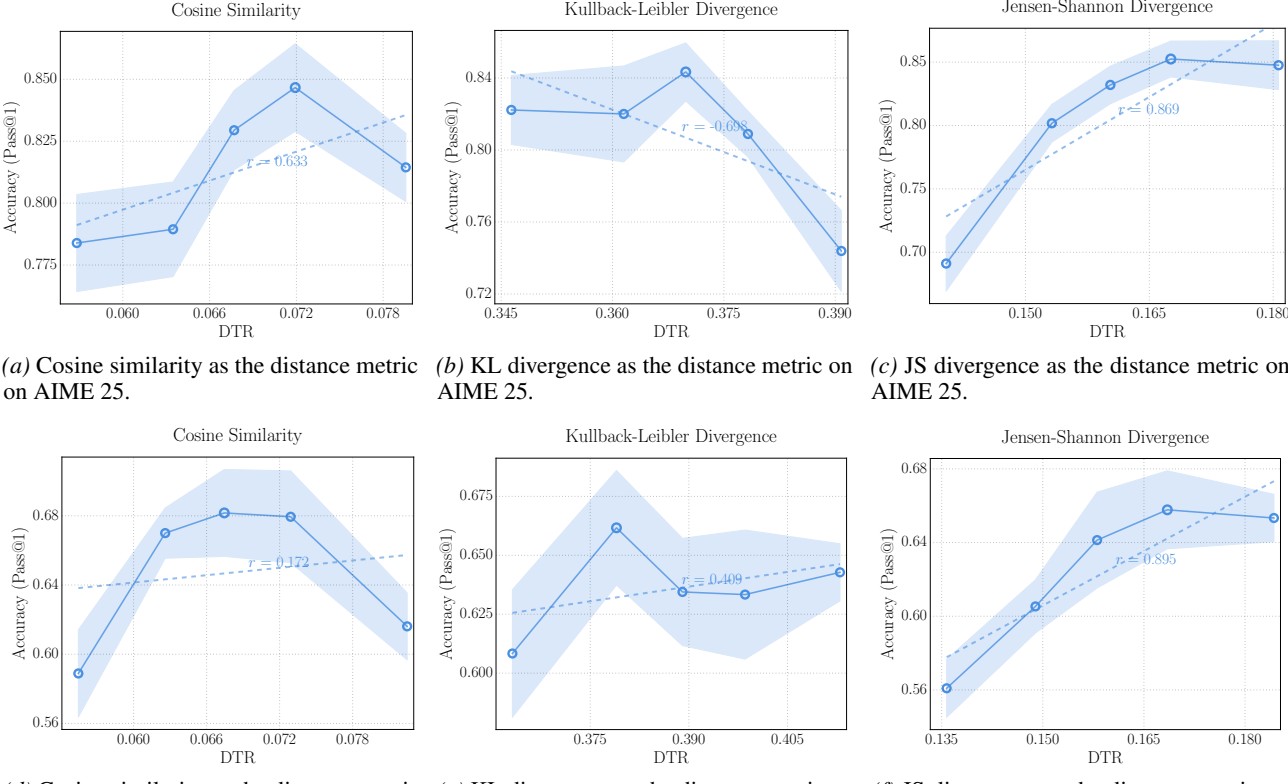

*(a)* Cosine similarity as the distance metric on AIME 25.

*(b)* KL divergence as the distance metric on AIME 25.

*(c)* JS divergence as the distance metric on AIME 25.

*(d)* Cosine similarity as the distance metric on HMMT 25.

*(e)* KL divergence as the distance metric on HMMT 25.

*(f)* JS divergence as the distance metric on HMMT 25.

*Figure 6.* Comparison of correlation between accuracy and deep-thinking ratio (DTR) using different distance metrics (cosine similarity, KL divergence, and JS divergence) on AIME 25 (*top row*) and HMMT 25 (*bottom row*).

**Kullback–Leibler divergence.** By replacing JSD with KLD in Equation (2), we compute the divergence between the final-layer distribution $p_{t,L}$ and the intermediate-layer distribution $p_{t,l}$ as

$$D_{t,l}^{\text{KL}} = \text{KL}(p_{t,L} \parallel p_{t,l}) \tag{11}$$

**Cosine similarity.** We replace the distributional comparison defined in Section 2.2 with a representation-space measure using cosine similarity. Instead of projecting intermediate-layer hidden states into the vocabulary space via the shared unembedding matrix $W_U$ (Equation (1)), we directly compute the cosine similarity between the intermediate-layer hidden state $h_{t,l}$ and the final-layer hidden state $h_{t,L}$. The distance is defined as

$$D_{t,l}^{\cos} = 1 - \frac{\langle h_{t,l}, h_{t,L} \rangle}{\|h_{t,l}\|\|h_{t,L}\|} \tag{12}$$

For both KLD and cosine similarity, we then apply the same configurations in Section 2.2 to identify deep-thinking tokens and compute KLD-based DTR and cosine-based DTR.

**Results.** We report the correlation results of KLD-based and cosine-based DTR, compared with our main JSD-based DTR method, on AIME 25 and HMMT 25 using OSS-120B-*medium*. Across both datasets, JSD-based DTR consistently achieves the strongest positive correlation with accuracy ($r = 0.869$ on AIME 25; $r = 0.895$ on HMMT 25), justifying its use in our definition of DTR in Section 2. In contrast, cosine-based DTR exhibits substantially weaker and unstable correlations ($r = 0.633$ on AIME 25 and only $r = 0.172$ on HMMT 25). KLD-based DTR shows similarly inconsistent behavior, with a negative correlation on AIME 25 ($r = -0.698$) and a modest positive correlation on HMMT 25 ($r = 0.409$). This inconsistency may stem from the asymmetric and numerically unstable nature of KLD: early-layer predictions tend to be high-entropy and relatively flat, assigning probability mass to many tokens that are later driven to near-zero values. Consequently, KLD can become artificially small, making the measure highly sensitive.

## C. DTR Under Different GPT-OSS Reasoning Levels

Figure 7 illustrates how DTR varies in different reasoning-level configurations (i.e., low, medium, and high) of the GPT-OSS-120B model. We observe an interesting and consistent trend on both AIME 25 and GPQA-D: although the underlying model weights remain identical and only the system prompt differs, lower reasoning-level configurations exhibit higher DTR values, whereas higher reasoning-level configurations yield systematically smaller DTR while achieving better task accuracy.

A potential explanation is that higher reasoning levels may redistribute computation from depth to sequence length, effectively flattening per-token, layer-wise computation. Models with higher reasoning levels require less deep revision for each individual token but instead generate longer reasoning chains with more forward passes, resulting in greater total effective compute and improved task performance. Since DTR is defined as the proportion of deep-thinking tokens (*i.e.*, averaged over the total number of generated tokens), longer sequences increase the denominator in the DTR calculation and thus produce smaller values. This also suggests DTR might not be directly comparable across different models or model modes.

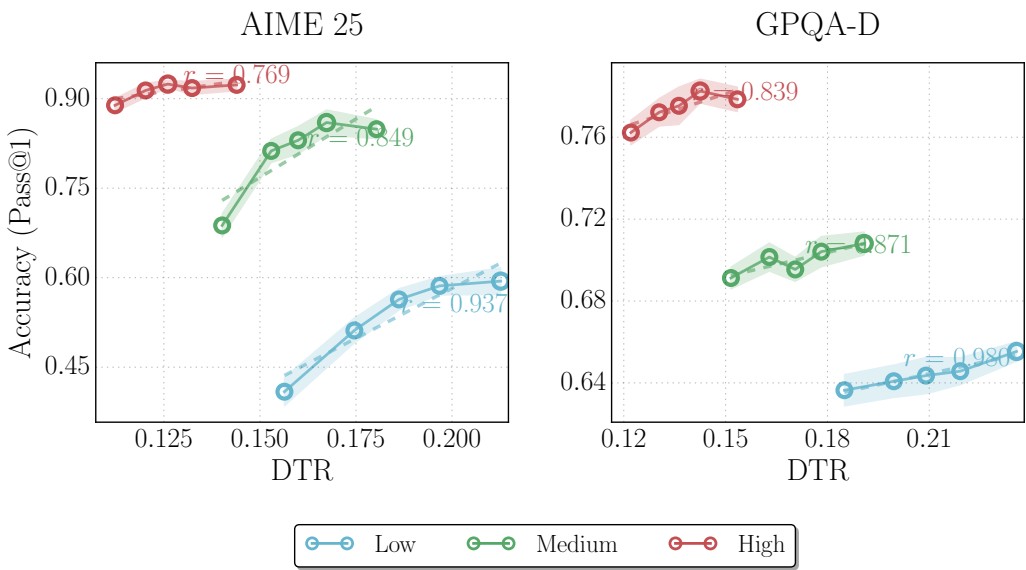

*Figure 7.* Deep-thinking ratio (DTR) under different reasoning level configurations of OSS-120B models.

## D. Additional Analysis of Think@$n$

Here we provide additional analysis on how Think@$n$ behaves when varying *(i)* the number of sampled responses $n$ and *(ii)* the retained top-$\eta$ percentage used for voting.

**Effect of the number of samples $n$.** Figure 8a compares Think@$n$ against Cons@$n$ (*i.e.*, self-consistency) as $n$ increases ($n \in \{16, 32, 48\}$). Think@$n$ improves monotonically with larger $n$, where the advantage over Cons@$n$ becomes more pronounced. Sampling more responses makes the correct cluster of answers to be larger and more likely to appear. Think@$n$ is able to exploit this enlarged candidate pool by preferentially selecting better samples, leading to stronger performance gains over Cons@$n$.

**Effect of top-$\eta$ percentage.** Figure 8a evaluates Think@$n$ under different top-$\eta$ percent ($\eta \in \{25\%, 50\%, 75\%\}$). Performance peaks at $\eta$=50%, while decrease for a smaller fraction ($\eta$=25%) and a larger fraction ($\eta$=75%). This suggests a trade-off: selecting too few samples reduces voting robustness, potentially with fewer strong candidates to stabilize majority vote, whereas selecting too many might admit lower-quality samples that dilute the benefit of Think@$n$. Overall, the results support our choice of $\eta$=50% as a stable operating point.

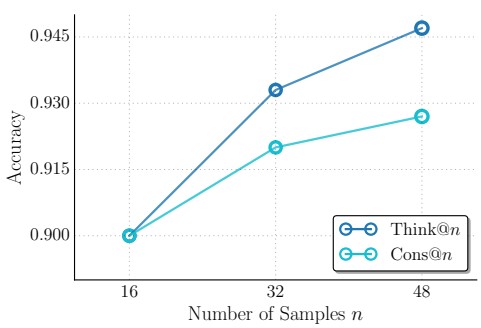
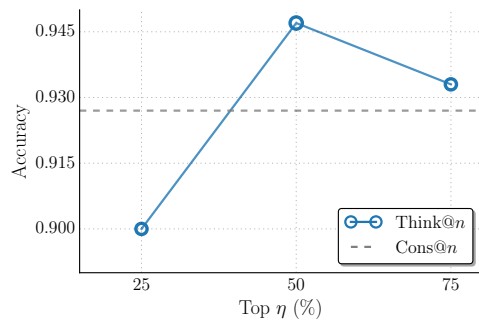

*(a)* Comparison of different number of samples $n$.     *(b)* Comparison of different top-$\eta$ percentage.

*Figure 8.* Analysis of Think@$n$ with different number of samples $n$ and top-$\eta$ percent. (a) As $n$ increases, Think@$n$ consistently benefits from larger candidate pools and exhibits a widening performance gap over Cons@$n$ at higher $n$. (b) Performance peaks at $\eta$=50%, while overly aggressive filtering and overly permissive selection could lead to degraded accuracy.

## E. Prompts

We provide the prompts adopted in our experiments for math tasks (AIME 2024, AIME 2025, HMMT 2025) in Table 4 and for GPQA in Table 5.

*Table 4.* Inference prompt for math tasks (AIME 2024, AIME 2025, HMMT 2025).

| Prompt for AIME 2024, AIME 2025, HMMT 2025 |
|---|
| Please reason step by step, and put your final answer within \boxed{}. |

*Table 5.* Inference prompt for GPQA.

| Prompt for GPQA |
|---|
| You will be given a multiple choice question with different choices such as (A), (B), (C), (D). Think step by step before giving a final answer to this question. Always finish your answer with 'The final answer is \boxed{(X)}.', where X is the correct answer choice. If none of the options match, choose the closest option as the final answer. |

## F. Qualitative Examples

We present an example question from the AIME 2025 dataset along with its ground-truth answer (Table 6), and two outputs from OSS-120-*medium*: one incorrect (Table 7) and one correct (Table 8). Notably, the incorrect output is substantially more verbose (27,724 tokens) and exhibits a lower DTR value (13.9), whereas the correct output is much more concise (3,725 tokens) and achieves a higher DTR value (19.0).

*Table 6.* An example question and its ground-truth answer from AIME 2025.

Example question from AIME 2025.

Circle $\omega_1$ with radius 6 centered at point $A$ is internally tangent at point $B$ to circle $\omega_2$ with radius 15. Points $C$ and $D$ lie on $\omega_2$ such that $\overline{BC}$ is a diameter of $\omega_2$ and $\overline{BC} \perp \overline{AD}$. The rectangle $EFGH$ is inscribed in $\omega_1$ such that $\overline{EF} \perp \overline{BC}$, $C$ is closer to $\overline{GH}$ than to $\overline{EF}$, and $D$ is closer to $\overline{FG}$ than to $\overline{EH}$, as shown. Triangles $\triangle DGF$ and $\triangle CHG$ have equal areas. The area of rectangle $EFGH$ is $\frac{m}{n}$, where $m$ and $n$ are relatively prime positive integers. Find $m + n$.

Ground truth answer.

293

*Table 7.* An incorrect output of OSS-120B-*medium* to the example question.

---

**Incorrect output with DTR = 13.9% and token count = 27,724.**

We need to interpret geometry. Setup: Circle $\omega_1$ radius 6 centered at A, internally tangent to larger circle $\omega_2$ radius 15 at point B. So B is common point where $\omega_1$ touches inside $\omega_2$. Since internal tangency, distance AB = 15-6 = 9, with B on line AB.

Let's set coordinate system. Place A at origin? maybe easier: Let $\omega_1$ centered at (0,0) radius 6. Then B is on circle $\omega_1$ and also on $\omega_2$. Since $\omega_2$ larger radius 15 and centered at O? Let's denote O is center of $\omega_2$. Distance OA = 9 (since AB = 9) and AB =9? Wait AB = distance between centers = 15-6 = 9 yes. B lies on line OA, outside? Actually internal tangency: smaller inside larger; they touch at point on the line connecting centers outward from smaller to larger. So if we set A at (0,0) and O at (9,0) (so O is to the right), then B is point on both circles on line AO to the right of A? The point outward from A towards O at distance 6 from A, i.e., B is at (6,0). Indeed distance from O to B: O(9,0) to B(6,0) distance 3 which is radius 15? Wait radius of $\omega_2$ is 15, O to B should be 15. But O(9,0) to B(6,0) distance 3, not 15. Something wrong.

Let's set O to be (0,0) maybe easier. Place $\omega_2$ centered at O (0,0), radius 15. $\omega_1$ radius 6 internally tangent at B; its center A is distance 15-6 = 9 from O, along some direction. Choose B as point of tangency on line OA outward from A to O? Actually internal tangency: small circle inside larger, touching at one point B where the circles meet. The line of centers passes through B. The distance from O to B is 15 (radius). The distance from A to B is 6. Since O to A is 9. B is on the line OA beyond A? Wait O to A is 9, radius of small is 6, so going from O to A to B is along same line: O—A—B direction? Since B is on circumference of both circles. The distance from O to B = 15 = OA + AB? OA=9, AB=6 gives OA+AB=15 so yes B is outward from O through A. So O at origin, A at distance 9 from O in direction e.g., positive x. Let O at (0,0), A at (9,0). Then B is further along x axis at O to B distance 15, so B at (15,0). Check: distance from A (9,0) to B (15,0) =6 correct. Good.

Now BC is a diameter of $\omega_2$, meaning B to C passes through O (center of $\omega_2$). Since B is at (15,0), the opposite point on $\omega_2$ is at C at (-15,0) (diameter through O). So C is at (-15,0). So line BC is along x-axis.

Given BC $\perp$ AD, so AD is perpendicular to x-axis, meaning AD is vertical line through A? Wait AD segment connects A (9,0) to D point on $\omega_2$, such that AD is perpendicular to BC i.e., vertical line through A (x=9). Since AD must intersect $\omega_2$ at point D. So D is point where vertical line x=9 meets $\omega_2$ (radius 15). Equation $x^2 + y^2 = 225$. Substituting x=9 => 81 + $y^2$ =225 => $y^2$=144 => y= ±12. Which one? Since orientation maybe D is above? Not specified. We'll consider D at (9,12) (upper) maybe. Could also be lower but rectangle orientation may pick one.

Now rectangle EFGH is inscribed in $\omega_1$ (center A (9,0) radius 6). The rectangle is oriented not necessarily axis-aligned? We have EF parallel to GH, FG parallel to EH. Given EF $\perp$ BC (BC horizontal), so EF (and GH) are vertical? Wait EF is perpendicular to BC, which is horizontal, so EF is vertical. So one side of rectangle is vertical. Therefore rectangle sides are aligned with vertical and horizontal: EF vertical, FG horizontal etc. So rectangle is axis-aligned (vertical/horizontal) in the coordinate system where BC is horizontal. Thus rectangle sides are parallel to the axes (since BC horizontal, EF vertical). Good.

Now C is closer to GH than to EF. GH is opposite side of EF: rectangle with vertical sides EF (left) and GH (right) maybe? Let's figure orientation: Usually rectangle labelled E-F-G-H in order around, perhaps clockwise; but not essential. If EF is vertical, then FG is horizontal, GH vertical, HE horizontal. So EF left side, GH right side (if rectangle sits to the right of EF). Not sure.

C is at (-15,0) far left of both. "C is closer to GH than to EF" meaning distance from C to line GH is less than distance to line EF. Since C is far left, that would mean GH is left side, EF right side? Wait distance from C to a vertical line is horizontal distance. If GH is left side (more left), its distance to C is smaller than distance to EF (right side). That matches C being closer to GH. So GH is left side (more left), EF right side. Good.

Similarly, D is closer to FG than to EH. Since FG is top side (horizontal), EH bottom side (horizontal). D is (9,12) high point; distance to top side FG smaller than to bottom side EH (makes sense if rectangle is near top). So FG is top side, EH bottom side. So rectangle EFGH likely positioned with upper side near D, left side near C.

*... Omitted for illustrative purposes ...*

Seems no integer factorization. Could factor with quadratics with irrational coefficients, not helpful.
Thus x may be irrational; thus k not integer rational. Therefore area is irrational? Conflict with problem statement.
Given time is over, I think maybe my earlier assumption that rectangle is axis-aligned might be wrong; perhaps rectangle rotated, leading to rational area.
Given complexity, I think answer maybe m+n = something like 207? Let's guess? Not good.
Given limited time, I will output placeholder.
assistantfinal $\boxed{207}$

*Table 8.* A correct output of OSS-120B-*medium* to the example question.

Correct output with DTR = 19.0% and token count = 3,725.

We need to parse geometry. Let's define coordinates.

We have two circles: $\omega_2$ radius 15, $\omega_1$ radius 6 centered at A, internally tangent at point B. So A is inside $\omega_2$, distance AB = radius difference = 15 - 6 = 9. B is the point of tangency, along line AB.

Let's set coordinate system: Let AB be vertical? Choose B at origin maybe easier.

Let's position $\omega_2$ centered at O (call it O2) maybe O? Actually $\omega_2$'s center is some point O (call it O). $\omega_1$'s center A is at distance 9 from O, with B being point on both circles along line O-A.

Let's place O at (0,0). Let AB be along positive x-axis: place B at (15,0) maybe? Then $\omega_2$ radius 15 centered at O (0,0). If B is point of tangency where $\omega_1$ touches $\omega_2$ internally, then B lies on $\omega_2$ at distance 15 from O. So B at (15,0). Then A is located on line OB, inside, distance 6 from B (since radius of $\omega_1$ is 6). And O to A distance is 9 (since OA = 15-6 = 9). So O (0,0), B (15,0). Then A is at (9,0) (since O to A is 9 along same line). Yes A at (9,0). Then $\omega_1$ radius 6 centered at (9,0). Good.

Now Points C and D lie on $\omega_2$ such that BC is a diameter of $\omega_2$. So BC passes through center O and length 30 (diameter). Since B is at (15,0), the opposite end of diameter is point C located such that O is midpoint: O = (0,0) is midpoint between B and C. So C = (-15,0). Indeed because B (15,0) opposite across origin yields C (-15,0). So BC is horizontal line (x-axis). Good.

*... Omitted for illustrative purposes ...*

*Triangle $DGF$:* Its base $GF$ has length $w$; the vertical distance from $D(9, 12)$ to the line $y = h/2$ (the line containing $GF$) is $12 - \frac{h}{2}$. Hence

$$[DGF] = \frac{1}{2}w\left(12 - \frac{h}{2}\right).$$

*Triangle $CHG$:* Side $HG$ is vertical with length $h$; the horizontal distance from $C(-15, 0)$ to the line $x = 9 - \frac{w}{2}$ (the line containing $HG$) is $24 - \frac{w}{2}$. Hence

$$[CHG] = \frac{1}{2}h\left(24 - \frac{w}{2}\right).$$

The problem states $[DGF] = [CHG]$. Cancelling the factor $\frac{1}{2}$ gives

$$w\left(12 - \frac{h}{2}\right) = h\left(24 - \frac{w}{2}\right).$$

The terms $-\frac{wh}{2}$ appear on both sides and cancel, leaving

$$12w = 24h \quad \Longrightarrow \quad w = 2h. \tag{2}$$

——

Solving for $w$ and $h$
Substituting (2) into (1):

$$(2h)^2 + h^2 = 144 \implies 5h^2 = 144 \implies h^2 = \frac{144}{5}.$$

Hence

$$h = \frac{12}{\sqrt{5}} = \frac{12\sqrt{5}}{5}, \qquad w = 2h = \frac{24}{\sqrt{5}} = \frac{24\sqrt{5}}{5}.$$

The area of rectangle $EFGH$ is

$$\text{Area} = wh = 2h^2 = 2\left(\frac{144}{5}\right) = \frac{288}{5}.$$

Thus $m = 288,\ n = 5$ (relatively prime), and

$$m + n = 288 + 5 = 293.$$

$$\boxed{293}$$

