# OpenReview forum: "Think Deep, Not Just Long: Measuring LLM Reasoning Effort via Deep-Thinking Tokens"
_ICML.cc/2026/Conference — ICML 2026 regular_

### Official Review · Reviewer_kdem · 2026-02-21

**Soundness:** 3
**Presentation:** 3
**Significance:** 3
**Originality:** 3
**Overall Recommendation:** 5
**Confidence:** 3

**Summary:**

This paper introduces deep-thinking ratio, a new metric for calibrating of inference-time thinking effort. It counts deep-thinking tokens, where internal predictions undergo significant revisions in deeper model layers before model convergence. Across four math and scientific benchmarks and a set of reasoning models, this new metric shows a robust and consistent positive correlation with accuracy. Based on this metric, this paper also proposes a test-time scaling strategy Think@n, which prioritizes samples with high deep-thinking ratio. This strategy reduces inference cost by enabling early rejection of non-promising generations.

**Compliance With Llm Reviewing Policy:**

Affirmed.

**Final Justification:**

My comments have been addressed by the rebuttal.

I will hold my score cause it is already high.

**Key Questions For Authors:**

Q. It might beneficial to study DTS performance on small models. Current Table1 study seems to on relative large models.

Q. It might be beneficial to study whether the DTS has shift after model quantization.

Q. Is it possible to use DTR for improving the inference accuracy, since it can measure which part of tokens needs more internal processes.

**Limitations:**

No.

I'm not aware of any negative societal impact here.

**Strengths And Weaknesses:**

S1. This paper studies an important problem in LLM reasoning — using too many tokens but still fails from time to time.

S2. The proposed metric deep-thinking ratio has some advantages over using the naive output length or total tokens, by counting only the tokens that reflects more thinking efforts internally.

S3. The metric defined based on Jensen-Shannon divergence seems to be sophisticated, giving the degree of how much the inner layer has been updated.

S4. The experiments indicates the effectiveness of the metric.

W1. The derived metric involves extra computation cost, which might be significant on large models. This paper does not include relevant study.

W2. The DTS metric has relevance with inference metrics in Table1, but this does not indicate causality from DTS to others.

---

> ### Author Rebuttal · Authors · 2026-03-31
>
> Appreciate the positive comments and thoughtful suggestions! The link to additional results is [here](https://anonymous.4open.science/r/ICML-Submission-29727-3B1F/Reviewer%20kdem.pdf).
>
> > W1. The derived metric involves extra computation cost, which might be significant on large models. This paper does not include relevant study.
>
> We profile DTR overhead in Table 1. The results suggest a reduction of throughput by about 42%. Note that our current implementation collects all intermediate hidden states during the forward pass and applies the unembedding matrix afterwards, introducing memory overhead that shows down inference. A more efficient implementation—applying the unembedding layer directly within the forward pass—would directly reduce memory usage and speedup generation while preserving the same results. We view this as an engineering optimization deferred to future deployment-oriented work; as a first attempt, we focus on establishing the validity of the DTR signal.
>
> > W2. The DTS metric has relevance with inference metrics in Table1, but this does not indicate causality from DTS to others.
>
> We agree that correlation does not imply causality, and we do not claim that DTR causes accuracy gains; we will revise any wording that may suggest otherwise. The Think@n results show that DTR offers useful signals to improve accuracy, which we hope provides additional support that DTR captures meaningful information beyond spurious correlation.
>
> > Q. It might beneficial to study DTS performance on small models. Current Table1 study seems to on relative large models.
>
> We have added results in Table 4 and Figure 6, for a smaller model, Qwen3-4B-Thinking. The results show that DTR remains effective and achieves the best overall Pareto-optimal performance.
>
> > Q. It might be beneficial to study whether the DTS has shift after model quantization.
>
> DTR performs effectively under quantization, as evidenced by the results of GPT-OSS series models, which are [natively MXFP4-quantized](https://openai.com/index/introducing-gpt-oss/). We additionally experiment on DeepSeek-R1-Distill-Llama-70B-FP8-dynamic and provides results in Table 5, where DTR achieves a correlation of 0.699 — a slight reduction from the unquantized version, while still strongly positive and tracks closely with the overall average DTR correlation in our paper's Table 1.
>
> > Q. Is it possible to use DTR for improving the inference accuracy, since it can measure which part of tokens needs more internal processes.
>
> We see two promising directions for leveraging DTR to improve inference accuracy. First, our Think@n already leverages DTR at the sequence level to improve accuracy via preferential selection. Second, a token-level extension would be to adaptively allocate extra computation or trigger resampling for promoting deep-thinking tokens when DTR monitoring shows diminishing returns—which we consider an exciting application in future!

---

> > ### Author Rebuttal · Reviewer_kdem · 2026-03-31
> >
> > My comments have been addressed. Thanks for the clarification.
> >
> > I will hold my score casue it is already high.

---

### Official Review · Reviewer_HZds · 2026-03-08

**Soundness:** 3
**Presentation:** 4
**Significance:** 3
**Originality:** 4
**Overall Recommendation:** 5
**Confidence:** 4

**Summary:**

They show that the proportion of the response that uses deep thinking has a stronger correlation with accuracy than other existing methods, such as reasoning trace length or uncertainty. They identify the deep-thinking ration (DTR) metric, a novel contribution to the subfield. They introduce a test-time intervention called Think@n that lowers inference cost and increases metrics like self-consistency.

**Compliance With Llm Reviewing Policy:**

Affirmed.

**Final Justification:**

Issues resolved; maintaining high accept score

**Key Questions For Authors:**

1. In addition to computing r^2 values, can you conduct a hypothesis test to show that your metric holds statistically significant, predictive signal?
2. Can you control for the lexical properties of your outputs? Are there certain types of words (ie. pronouns vs. specific proper nouns) that are also correlated with a high “deep thinking” ratio?
3. How sensitive are the main conclusions to alternative analysis choices beyond Pearson correlation and 5-bin aggregation, such as Spearman correlation, per-instance ranking metrics, or simple regression models that jointly include DTR, length, and confidence?

**Limitations:**

yes

**Strengths And Weaknesses:**

**Soundness**

Soundness strengths:

- This sentence was very clarifying for me: “Note that, our proposed method does not imply that early-settling tokens are suboptimal; rather, it provides a depth-wise characterization of inference-time thinking effort that complements the surface-level token length measure.” Is there a way to make this idea shine more in the abstract or introduction?
- The authors compare their new metric with common baselines like token count, reverse token count, and uncertainty measurements.
- Your results span 8 models, which is an appropriate number.
- This work reproduces recent results that overthinking is correlated with lower accuracy values.
- There is adequate robustness testing for choosing values of g and p.
- It was excellent to compare the inference costs of each of the baseline strategies that you considered.

Soundness concerns:

- The r^2 tests are limited. You might want to augment the types of analyses that you do to include hypothesis tests that show whether your metric holds statistically significant, predictive signal in accuracy. At the least, include some measurement of confidence intervals around your results.
- Can you control for the lexical properties of your outputs? I’d suspect that certain types of words (ie. pronouns vs. specific proper nouns) are also correlated with a high “deep thinking” ratio. I have concerns that the ratio captures aspects of model output that are not properties of the model’s reasoning traces.
- There is almost no Appendix. Consider adding implementation details and a detailed specification of your algorithm to the appendix.

**Presentation**

Presentation strengths:

- Table 1 was clear and easy to read.
- The paper holistically was notably well-written!

Presentation weaknesses:

- I found some of your choices of the language of the terms to be slightly too anthropomorphic (ie. ‘additional thinking’ and ‘internal thinking’). Are there other options that might work?
- In Figure 2, include some intuitive pointers (ie. ‘darker’ means not yet converged). It would also be helpful to sort column-wise, or at least group by category (ie. operation tokens vs. near-the-end tokens).
- The figures were all quite aesthetic.
- Minor nit: In this sentence, “Following standard practice in natural overthinking analyses …“, you cite only one (very recent) paper. As such, I would avoid calling it standard practice.
- To avoid complains about not testing the frontier models (ie. OpenAI, Anthropic, etc.), make sure you mention that your coverage is limited to open-sourced models because your method relies on internal measurements.
- In Table 1, include categorization indicators for which of the measurements are length-based, confidence-based, etc.
- Wording issue: “Varying g has more impacts the correlation”
- In Figure 4, make sure you indicate which values you settled on after hyperparameter testing and why.
- Wording issue: “exhibit more robust positive correlations reflecting the accuracy.”



**Significance**:
Significance strengths:

- Overall, an important concept investigated by the manuscript is whether inference-time reasoning effort can be measured more faithfully than by raw CoT length alone. That is a meaningful problem for the field because token length is widely used as a proxy in both analysis and test-time scaling, despite growing evidence that it is often misleading.

Significance weaknesses:

- The practical significance is somewhat constrained by the requirement for access to intermediate hidden states and layerwise vocabulary projections. This makes the method most applicable to open-weight/open-internal models, and less immediately useful for many high-impact production APIs where such internals are inaccessible.



**Originality**

Originality strengths:

- Most prior work focused on improving performance using the length of the reasoning output. It does not involve “deep thinking”, as defined by this paper.
- The “deep thinking” metric is novel. They project intermediate hidden states onto the output vocabulary. They define “deep thinking” as when the vocabulary only converges / settles in later layers. This idea is not represented in the literature and, given the empirical evidence provided in the paper, seems to be the best internal predictive metric of accuracy that we have so far.
- There has been a lot of focus in the research community on output-level predictive measurements of accuracy. This work provides a novel *internal* metric that works.

---

> ### Author Rebuttal · Authors · 2026-03-31
>
> Thanks for the positive feedback and constructive suggestions! The link to additional results is [here](https://anonymous.4open.science/r/ICML-Submission-29727-3B1F/Reviewer%20HZds.pdf).
>
> > … You might want to augment the types of analyses that you do to include hypothesis tests that show whether your metric holds statistically significant …
>
> > … can you conduct a hypothesis test to show that your metric holds statistically significant, predictive signal?
>
> We conducted a one-sided t-test for each of the 32 model-benchmark settings and observe DTR records a statistically significant positive correlation (p < 0.05) in 28 out of 32 settings, suggesting that its association with task accuracy holds statistically significant rather than chance. We will augment the results in the revised manuscript.
>
> > … I’d suspect that certain types of words (ie. pronouns vs. specific proper nouns) are also correlated with a high “deep thinking” ratio …
>
> > Can you control for the lexical properties of your outputs? …
>
> We agree that lexical properties is a relevant factor, as illustrated in the paper’s Figure 2, functional and templated words tend to settle at shallow layers. However, we do not assume that only words semantically related to reasoning are classified as deep-thinking tokens. Rather, any token that undergoes sustained layer-wise revision before converging is counted, regardless of its lexical category. We agree that a future analysis categorizing deep-thinking tokens by lexical properties or part-of-speech tags would be interesting and could provide additional interpretability!
>
> > … Consider adding implementation details and a detailed specification of your algorithm to the appendix.
>
> > … choices of the language of the terms to be slightly too anthropomorphic …
>
> > In Figure 2, include some intuitive pointers … be helpful to sort column-wise, or at least group by category …
>
> > … “Following standard practice in natural overthinking analyses …“, you cite only one (very recent) paper … avoid calling it standard practice.
>
> > In Table 1, include categorization indicators for which of the measurements are length-based, confidence-based, etc.
>
> > Wording issue: “Varying g has more impacts the correlation”
>
> > In Figure 4, make sure you indicate which values you settled on after hyperparameter testing and why.
>
> > Wording issue: “exhibit more robust positive correlations reflecting the accuracy.”
>
> Thanks for the detailed presentational feedback! We will incorporate the mentioned figure, wording, grammar, and descriptive suggestions in the revised version.
>
> > … make sure you mention that your coverage is limited to open-sourced models because your method relies on internal measurements.
> > … constrained by the requirement for access to intermediate hidden states and layerwise vocabulary projections. This makes the method most applicable to open-weight/open-internal models …
>
> Thanks for the suggestion – we will explicitly discuss the requirement of white-box access and DTR's applicable scope in the revision.
>
> > How sensitive are the main conclusions to alternative analysis choices beyond Pearson correlation and 5-bin aggregation, such as Spearman correlation, per-instance ranking metrics, or simple regression models that jointly include DTR, length, and confidence?
>
>
> We have added results in Table 6 and 7 showing our conclusions hold under alternative analysis choices. For alternative correlation metrics: DTR achieves a Spearman correlation of 0.900, slightly exceeding its Pearson counterpart, confirming the robustness of our findings under different correlation metrics. For alternative aggregation granularities: DTR maintains strong positive correlations at 10 bins (r = 0.784) and per-instance ranking (r = 0.747). The slight decrease at finer granularity is expected given the binary nature of per-instance accuracy. For training joint regression models: as a first attempt, we focus on establishing the validity of DTR as a standalone, training-free, zero-shot metric. We agree that combining length and confidence via learned models is an interesting practical direction, which we defer to future deployment-oriented work.

---

> > ### Author Rebuttal · Reviewer_HZds · 2026-04-01
> >
> > Issues fully resolved, maintaining my Accept score.

---

### Official Review · Reviewer_aD2n · 2026-03-09

**Soundness:** 4
**Presentation:** 4
**Significance:** 4
**Originality:** 4
**Overall Recommendation:** 6
**Confidence:** 4

**Summary:**

Overall, an important concept investigated by the manuscript is how to faithfully measure inference-time reasoning effort beyond surface-level token counts. Overall, the authors analyze a core challenge in test-time compute scaling: the disconnect between generation length and actual reasoning quality.

**Compliance With Llm Reviewing Policy:**

Affirmed.

**Key Questions For Authors:**

## Suggested Additional References

* "Layer by Layer: Uncovering Hidden Representations in Language Models": This work demonstrates the importance of intermediate-layer representations, which is directly relevant to the layer-wise analysis perspective adopted in this paper and could enrich the discussion on how models utilize their depth.
* "How Chain-of-Thought Works? Tracing Information Flow from Decoding, Projection, and Activation": This work analyzes the CoT mechanism from an information flow perspective, covering the dimensions of projection and activation, which methodologically complements this paper's use of logit lens projection to trace the reasoning process.

## Questions and Recommendations

* Although Figure 2 provides an illustrative example, I would appreciate seeing aggregate statistics. Specifically: (1) a line plot showing how JSD varies across layers (x-axis: layer index, y-axis: average JSD), which would reveal whether convergence predominantly occurs at lower, middle, or upper layers; (2) a histogram of the settling depth distribution, showing what proportion of tokens converge at each layer. This would clarify whether deep-thinking tokens are predominantly produced by a specific subset of layers.
* Why was distribution-level JSD chosen over token-level trajectory analysis? For instance, one could directly observe the layer-wise evolution of the top-1 predicted token (e.g., semantic jumps such as "hello" → "8" → "13").

**Limitations:**

yes

**Strengths And Weaknesses:**

## Strength:

* Novel and important research motivation: The paper identifies a prevalent issue that token length does not equate to reasoning quality, and proposes a mechanism-level alternative. The research perspective is refreshing.
* Sound method design with clear physical intuition: DTR is grounded in the layer-wise convergence behavior of internal predictive distributions, offering clear interpretability without relying on external annotations or auxiliary models.
* High writing quality: The paper is logically coherent and structurally compact, progressing seamlessly from motivation to method to experiments to application, making it easy to follow.
* Comprehensive experiments with strong generalizability: The evaluation covers 3 model families (GPT-OSS, DeepSeek-R1, Qwen3), 8 model variants, and 4 benchmarks, with consistent results throughout. The hyperparameter sensitivity analysis (Section 3.2) further strengthens the credibility of the method.
* Clear practical value: The Think@n strategy directly translates DTR into an actionable inference acceleration scheme, with the approximately 50% cost reduction being of practical significance.

## Weakness:

Overall, I very much enjoyed reading this paper and look forward to the open-source release of the code and data.

---

> ### Author Rebuttal · Authors · 2026-03-31
>
> Appreciate the positive feedback and thoughtful comments! The link to additional results is [here](https://anonymous.4open.science/r/ICML-Submission-29727-3B1F/Reviewer%20aD2n.pdf).
>
> > "Layer by Layer: Uncovering Hidden Representations in Language Models" …
>
> > "How Chain-of-Thought Works? Tracing Information Flow from Decoding, Projection, and Activation" …
>
> Thank you for mentioning these relevant works — we will be sure to incorporate them into our discussion!
>
> > … I would appreciate seeing aggregate statistics. Specifically: (1) a line plot showing how JSD varies across layers (x-axis: layer index, y-axis: average JSD), which would reveal whether convergence predominantly occurs at lower, middle, or upper layers; (2) a histogram of the settling depth distribution, showing what proportion of tokens converge at each layer. This would clarify whether deep-thinking tokens are predominantly produced by a specific subset of layers.
>
> We provide the plots showing aggregation statistics in Figure 2-5, for AIME25 and GPQA-D, on GPT-OSS-120B (36 layers).
>
> (1) Mean JSD across layers. Both benchmarks show an interesting and consistent pattern: intermediate-layer distributions remain far from the final-layer distribution throughout roughly the first half of the network, before steeply converging around layers 16–20.
>
> (2) Settling depth distribution. The histogram also shows a broadly similar pattern on both benchmarks: deep-thinking tokens spread relatively uniformly across layers 31–35 within the late-settling regime, instead of clustering at a single layer.
>
> > Why was distribution-level JSD chosen over token-level trajectory analysis? For instance, one could directly observe the layer-wise evolution of the top-1 predicted token (e.g., semantic jumps such as "hello" → "8" → "13").
>
> We employ distribution-level JSD as it naturally aligns with the actual decoding process, where LLMs (especially reasoning models) sample stochastically from the full distribution rather than greedily. Additionally, top-1 token trajectories only provide sparse, lossy signals, potentially too coarse to meaningfully capture behavioral shifts across layers.

---

> > ### Author Rebuttal · Reviewer_aD2n · 2026-04-01
> >
> > Thanks for your figures, it is a very interesting work. I keep my score of 6.

---

### Official Review · Reviewer_44xx · 2026-03-12

**Soundness:** 3
**Presentation:** 3
**Significance:** 3
**Originality:** 3
**Overall Recommendation:** 5
**Confidence:** 4

**Summary:**

This paper argues that reasoning length (number of generated tokens) is an unreliable proxy for “reasoning effort”, and proposes a white-box metric based on *layerwise convergence* of token distributions. For each generated token, the authors project hidden states from each layer to the vocabulary distribution via the unembedding matrix, compute the Jensen–Shannon divergence (JSD) between each intermediate-layer distribution and the final-layer distribution, and define a token as a *deep-thinking token* if its distribution only becomes close to the final distribution at sufficiently late layers. The *Deep-Thinking Ratio (DTR)* is the fraction of deep-thinking tokens in a response. Across several reasoning/knowledge benchmarks and model families, DTR is reported to correlate with accuracy more consistently than output length. The paper further proposes a test-time scheme (Think@n) that uses a short prefix to estimate DTR, filters higher-DTR candidates among best-of-n samples, and aggregates/votes to achieve self-consistency-like accuracy with reduced token cost.

**Compliance With Llm Reviewing Policy:**

Affirmed.

**Final Justification:**

The rebuttal addressed my main concerns

**Key Questions For Authors:**

1. Does DTR remain predictive if you use a *tuned lens* / layerwise calibration instead of the raw unembedding? Does calibration reduce or explain the negative-correlation cases?
2. Can you analyze the *failure regimes* where DTR correlates negatively with accuracy (error types, hallucination vs arithmetic slips, etc.) and clarify when DTR should/should not be trusted?
3. How sensitive are the comparisons to *decoding settings* (temperature/top-p), especially since different models appear to use different decoding configs?
4. For Think@n, how does performance change with selection ratio and alternative aggregation schemes (top-k by DTR, DTR-weighted voting, DTR+confidence ranking)? Please provide full cost–accuracy curves.
5. What is the *actual overhead* of computing layerwise distributions/JSD? Can top-k approximations preserve the gains?

**Limitations:**

The paper should more explicitly state that the method requires white-box access to multi-layer activations/logits (thus not applicable to black-box APIs), quantify computational/storage overhead, and emphasize that DTR can be misleading in certain regimes (as evidenced by negative correlations), so it should be treated as an auxiliary signal rather than a universal proxy. These gaps (unexplained failure regimes and unquantified overhead) make it hard to assess whether DTR is a reliably deployable proxy beyond the reported settings.

**Strengths And Weaknesses:**

**Strengths**

- Timely and relevant: highlights the well-known mismatch between longer outputs and better reasoning, and provides an internal signal that often correlates better with accuracy than length.
- The metric is motivated by a plausible mechanistic intuition (late-layer “corrections” vs early stabilization), but the interpretability of layerwise distributions depends on how faithful the chosen lens is.
- Think@n is training-free and easy to integrate into existing best-of-n / self-consistency workflows, with promising cost–accuracy trade-offs in some settings.

---

**Weaknesses**

- The core measurement relies on an *uncalibrated logit lens* (same unembedding applied to all layers). Layerwise logits are known to be misaligned; without a tuned/learned lens, JSD changes may conflate reasoning dynamics with representation miscalibration, weakening interpretability and cross-model comparability.
- While DTR appears more consistent than length on average, it still exhibits non-trivial failure cases (including negative correlations) that are not analyzed in depth (e.g., error modes, decoding effects, or lens misalignment), weakening “general proxy” claims.
- Think@n is evaluated on a relatively narrow set of models/tasks, and the strategy space is under-explored (e.g., top-k by DTR, DTR-weighted voting, DTR combined with confidence/consistency signals, full curves over selection ratio).
- Practicality is limited to white-box access (multi-layer states/logits). The computational overhead (full-vocab JSD) and possible approximations (e.g., top-k) are not quantified in a deployment-relevant way.

---

> ### Author Rebuttal · Authors · 2026-03-31
>
> Thanks for the positive comments and constructive feedback! The link to added results is [here](https://anonymous.4open.science/r/ICML-Submission-29727-3B1F/Reviewer%2044xx.pdf).
>
> > … relies on an uncalibrated logit lens …
>
> > Does DTR remain predictive if you use a tuned lens … Does calibration reduce or explain the negative-correlation cases?
>
> Measuring native effort: DTR aims to measure the model's native computational effort, not to accelerate decoding. In complex reasoning, a model may surface a correct latent concept in early layers but continually revise it before finalization. A tuned lens would artificially short-circuit this process by decoding the concept early, hiding the depth-wise deliberation that DTR is designed to capture.
>
> Empirical sufficiency: DTR requires the relative separation of token complexities, not perfect early-layer probability calibration. As demonstrated in paper’s Figure 2, the untuned lens is sufficient for this: "easy" tokens naturally project through the final unembedding matrix at early layers, whereas heavy reasoning tokens remain unconverged until the final layers.
>
> Computational efficiency: Learning a tuned lens requires training model-specific affine transformations on auxiliary datasets. By relying on the untuned logit lens, DTR remains a training-free, zero-shot metric without introducing inference-stage training overhead.
>
> > … still exhibits non-trivial failure cases …
>
> > … analyze the failure regimes … clarify when DTR should/should not be trusted?
>
> We observe negative correlation in only 2 out of 32 settings, both involving smaller models (OSS-20B and Qwen3-30B). Focusing on Qwen3 on AIME 24, which has the more notable negative value, we identify two main failure modes in Table 3 example traces:
>
> Unproductive self-refinement. Models repeatedly exhibit self-refinement behaviors (e.g., "Wait”) and restart derivations without convergence, accumulating substantial computation that inflates DTR.
> Deep processing within a wrong reasoning frame. Models invest heavy effort into an incorrect strategy, producing high DTR in the process.
>
> We posit that DTR is less reliable for weaker models as both failure modes stem from insufficient model capacity. The lack of self-monitoring ability in weaker models could cause high DTR to reflect misdirected effort rather than effective reasoning.
>
> > Think@n … the strategy space is under-explored …
>
> > … how does performance change with selection ratio … aggregation schemes (top-k by DTR, DTR-weighted voting, DTR+confidence ranking)? …
>
> We provide results in Figure 1. As shown, selecting too few samples reduces voting robustness—potentially leaving too few strong candidates to stabilize the majority vote—whereas selecting too many may admit lower-quality samples that dilute the benefit of Think@n. Overall, the results support our choice of 50% as a stable operating point. Our original proposed Think@n (DTR top-50% with majority voting) outperforms DTR-weighted voting and DTR+Self-Certainty at similar cost. Note both DTR-weighted voting and DTR+Self-Certainty consistently outperform confidence-only Self-Certainty@n, underscoring the effectiveness of the DTR signal.
>
> > … computational overhead (full-vocab JSD) and possible approximations … not quantified …
>
> > What is the actual overhead … Can top-k approximations preserve the gains?
>
> We report DTR overhead in Table 1. Computing full-vocabulary JSD reduces throughput by about 42%. A top-40 approximation yields slightly lower throughput by about 47%, potentially due to the additional operation of selecting the top-40 logits from the final layer and mapping them to corresponding indices in the intermediate-layer logits before computing JSD.
>
> For the ease of implementation, we currently collect all intermediate-layer hidden states during the forward pass and apply the unembedding matrix afterwards, introducing memory overhead that slows down inference. A more efficient implementation—applying the unembedding layer directly within the forward pass—would directly reduce memory and speedup generation with the same results. We view this as an engineering optimization deferred to future deployment-oriented work; as a first attempt, we focus on establishing the validity of the DTR signal.
>
> > How sensitive are the comparisons to decoding settings …
>
> We report results in Table 2. DTR's correlation with accuracy remains consistently strong across temperature/top-p values, demonstrating that DTR generalizes robustly across decoding configs.
>
> > … requires white-box access …, quantify computational/storage overhead … can be misleading in certain regimes …
>
> We will explicitly state the requirement of white-box access in the revision and incorporate discussion on DTR's applicable scope.
>
> We’re more than happy to address any additional questions you might have. If our responses adequately address your concerns, we’d greatly appreciate your consideration in updating the score. Thanks again for your time!

---

> > ### Author Rebuttal · Reviewer_44xx · 2026-04-02
> >
> > My concerns have been addressed. I have raised my score.

---

### Decision · Program_Chairs · 2026-04-30

**Decision:**

Accept (regular)

**Comment:**

The paper tackles an important and timely question in LLM reasoning: how to measure inference-time reasoning effort more faithfully than by raw chain-of-thought length. The central proposal, Deep-Thinking Ratio (DTR), is well-motivated. By identifying tokens whose intermediate-layer predictions only stabilize late in the network, the paper offers a more mechanistic proxy for reasoning effort than surface-level token counts. Reviewers generally found this idea interesting, technically sound, and practically relevant, and they also viewed the accompanying Think@n strategy as a useful test-time application of the metric. Across the reviews, there was broad agreement that the paper is clearly written, experimentally substantial, and addresses a meaningful problem in test-time scaling for reasoning models.

The main concerns raised by reviewers centered on interpretability and scope rather than on fatal flaws. In particular, reviewers questioned the reliance on the raw logit lens, the computational overhead of computing DTR, the white-box requirement, the lack of deeper analysis of failure cases where DTR correlates negatively with accuracy, and whether additional statistical analyses or controls would strengthen the empirical claims. The authors’ rebuttal addressed these points substantively: they clarified the motivation for using the untuned lens, added analysis of negative-correlation regimes, reported overhead measurements, discussed robustness across decoding settings, added results on smaller and quantized models, and provided additional statistical evidence and alternative analyses.

I find the paper to be a solid accept. It introduces a distinctive and practically meaningful idea, supports it with a broad empirical study, and responds well to reviewer concerns.